# Equity in awareness and utilization of cervical cancer screening services among women of reproductive age in Uganda: Analysis of vertical equity using evidence from UDHS 2022

Geofrey Emesu[1]*, Elizabeth Ekirapa Kiracho[2], Alfred Jatho[3], Joseph Kagaayi[1], Aggrey David Mukose[1], Annette Kyomuhangi[1]

1 Department of Epidemiology and Biostatistics, Makerere University School of Public Health, Kampala, Uganda, 2 Department of Health policy and Planning, Makerere University School of Public Health, Kampala, Uganda, 3 Department of Community Cancer Services, Uganda Cancer Institute, Kampala, Uganda

* geofreyemesu25@gmail.com

## Abstract

### Background

Cervical cancer poses a severe public health burden in Uganda, which has one of the world's highest incidence rates. Despite commitments to Universal Health Coverage (UHC), screening utilization remains critically low and inequitable. This study assessed vertical equity in the awareness and utilization of cervical cancer screening services among women in Uganda, evaluating whether distribution aligns with differential need.

### Methods

We conducted a cross-sectional analysis of the 2022 Uganda Demographic and Health Survey (UDHS), including 18,251 women aged 15–49. The primary outcomes were self-reported screening utilization and awareness. Socioeconomic status was measured using the DHS wealth index. Equity was assessed using concentration curves and indices (CIs), with a positive CI indicating pro-rich inequality. P.value of 0.05 (95% confidence interval) was used to test for significance of study findings.

### Results

We found significant pro-rich inequity in both screening utilization (CI=0.125, p<0.000) and awareness (CI=0.178, p<0.000), demonstrating that wealthier women had a disproportionate advantage. The pro-rich inequality in utilization was more pronounced in the urban (CI=0.125) than rural (CI=0.049) areas. Awareness was distributed almost equitably in rural areas (CI=−0.007, p=0.165) but showed significant pro-rich inequality in urban settings (CI=0.014, p<0.016).

**Data availability statement:** In compliance with the PLOS Data Policy, which stipulates that authors cannot serve as the sole custodians of shared data, the underlying data for this study are available from permanent third-party repositories. The raw data are from the Uganda Demographic and Health Survey (UDHS) 2022, owned by the Uganda Bureau of Statistics (UBOS). Researchers can request access via The DHS Program https://dhsprogram.com or by contacting the UBOS registrar at statistics@ubos.org and the authors accessed the data under license hence prohibited from redistribution. To replicate the findings, all related analysis code and derived datasets are available from the corresponding author upon reasonable request.

**Funding:** The author(s) received no specific funding for this work.

**Competing interests:** Authors have no competing Interests.

## Conclusion

Uganda's cervical cancer screening services demonstrate significant vertical inequity, disproportionately favoring wealthier and urban women rather than being allocated according to greater need. To achieve elimination goals, deliberate policies must prioritize resource allocation and awareness for the poorer and rural women who bear the highest burden.

## Background

Cervical cancer represents a devastating and preventable public health burden across low- and middle-income countries, standing as the leading cause of gynecological cancer mortality globally and affecting 6.9% of all women [1]. Uganda faces an exceptionally severe crisis, bearing one of the world's highest burdens. The nation ranks among the top ten globally for cervical cancer incidence, estimated at 28.8 per 100,000 women annually, and second in Eastern Africa. This translates to approximately 6,413 new cases and 4,301 deaths reported each year [2]. This profound loss of life starkly contradicts the World Health Organization's core health system goals of equity, equality, and efficiency, highlighting a critical shortfall in achieving Universal Health Coverage —the global commitment ensuring all people access quality essential health services without enduring financial hardship.

In May 2018, the WHO launched a global effort to eliminate cervical cancer as a public health issue through a comprehensive strategy advocating for countries to meet the '90-70-90' targets by 2030: vaccinating 90% of girls against HPV by age 15, screening 70% of women with high-performance tests by ages 35 and 45, and ensuring that 90% of women with cervical pre-cancer or invasive cancer have access to treatment [3]. However progress has been slow or almost stagnated for most countries in Sub-Saharan Africa and by 2021, 133 countries reported slightly over 10% of women of reproductive age screened within the past 5 years [4]).

The core of this crisis lies in the alarmingly low and inequitable utilization of cervical cancer screening, For example in Uganda, Only an estimated 10% of eligible women have ever undergone cervical cancer screening with the national average obscuring stark and unacceptable disparities evidenced by screening rates plummeting to a mere 4.8% among rural women compared to 7% in urban settings [5]. This geographic inequity is particularly concerning given that rural women simultaneously experience a higher prevalence of infection with high-risk human papillomavirus at 18% compared to 11% in urban areas [6,7]—the primary causative agent of cervical cancer. The direct consequence of this low and uneven screening uptake is catastrophic with approximately 80% of Ugandan women diagnosed with cervical cancer at advanced stages of the disease, resulting in over 4,500 preventable deaths annually and a devastating mortality rate of 41.4 per 100,000 women [8].

Despite Uganda's formal adoption of the WHO UHC declaration and efforts to extend health services through a decentralized delivery system, cervical cancer screening remains erratic, opportunistic, and in many places entirely absent [9]. This

systemic failure is primarily attributed to a critical lack of resources and insufficient financial commitment. Consequently, screening utilization remains staggeringly low, entrenching profound disparities in access and leading to clear inequities in service utilization. While raising awareness is crucial—as evidence indicates women with gaps in knowledge and low perceived risk are significantly less likely to utilize services even when available—the fundamental barriers are structural and systemic [10,11].

This study therefore directly addresses a critical gap in the existing evidence base. Few prior investigations have specifically focused on the inequities in access to and utilization of cervical cancer screening services between different population groups within Uganda. Therefore, this research aims to fill this void by rigorously examining disparities among rural and urban residents and across socioeconomic strata. The justification for this focused inquiry rests on several key pillars. Firstly, the study is intrinsically grounded in Uganda's commitment to UHC, explicitly exploring health service utilization and access in relation to equity within this vital policy framework. Secondly, the findings possess direct and significant potential to inform the formulation and optimization of health policies specifically aimed at realizing UHC objectives for cervical cancer prevention and control. Finally, and most significantly, this research provides a crucial analysis of equity differences in health access and utilization between urban and rural residents—a critically under-explored dimension within the current Ugandan literature on cervical cancer.

Most importantly, the analysis simultaneously examined both demand-side (awareness creation) and supply-side (service utilization) dimensions of cervical cancer screening under a vertical equity framework, assessing whether current patterns across rural/urban populations and wealth quintiles are commensurate with differential biological need and socioeconomic disadvantage. Understanding these specific dimensions of inequity is fundamental to designing effective, targeted interventions that align with UHC principles and hold the potential to ultimately reduce Uganda's devastating cervical cancer burden. This research offers policymakers the granular data necessary to ensure lifesaving screening reaches those most vulnerable and most in need.

## Methods

### Study design

This study employed a cross-sectional design guided explicitly by the principle of vertical equity to assess and quantify disparities in cervical cancer screening utilization and awareness among women of reproductive age in Uganda, utilizing the 2022 Uganda Demographic and Health Survey.

### Vertical equity framework

Equity in health refers to the absence of systematic disparities in health (or major social determinants of health) between groups with different levels of underlying social advantage/disadvantage like wealth, power, or prestige. Inequity can be described as those inequalities that are unfair and it systematically puts groups of people who are already socially disadvantaged (for example, by virtue of being poor, female, and/or members of a disenfranchised racial, ethnic or religious group) at further disadvantage with respect to health [12].

Vertical equity is a fundamental tenet of distributive justice within health economics and policy, it asserts that individuals or groups with differing levels of need and underlying disadvantage should receive appropriately differentiated levels of resources or services to achieve equitable health outcomes [13]. It necessitates allocating resources disproportionately towards populations facing greater burdens of disease and/or systematic barriers to access, moving beyond horizontal equity's focus on equal treatment for equal need.

The framework below (Fig 1) on UHC and the intersection of need and provision of services highlights the importance of the vertical equity approach in achieving equity in access and utilization of services driven by need. The need can either be influenced by demand or supply forces in health care delivery.

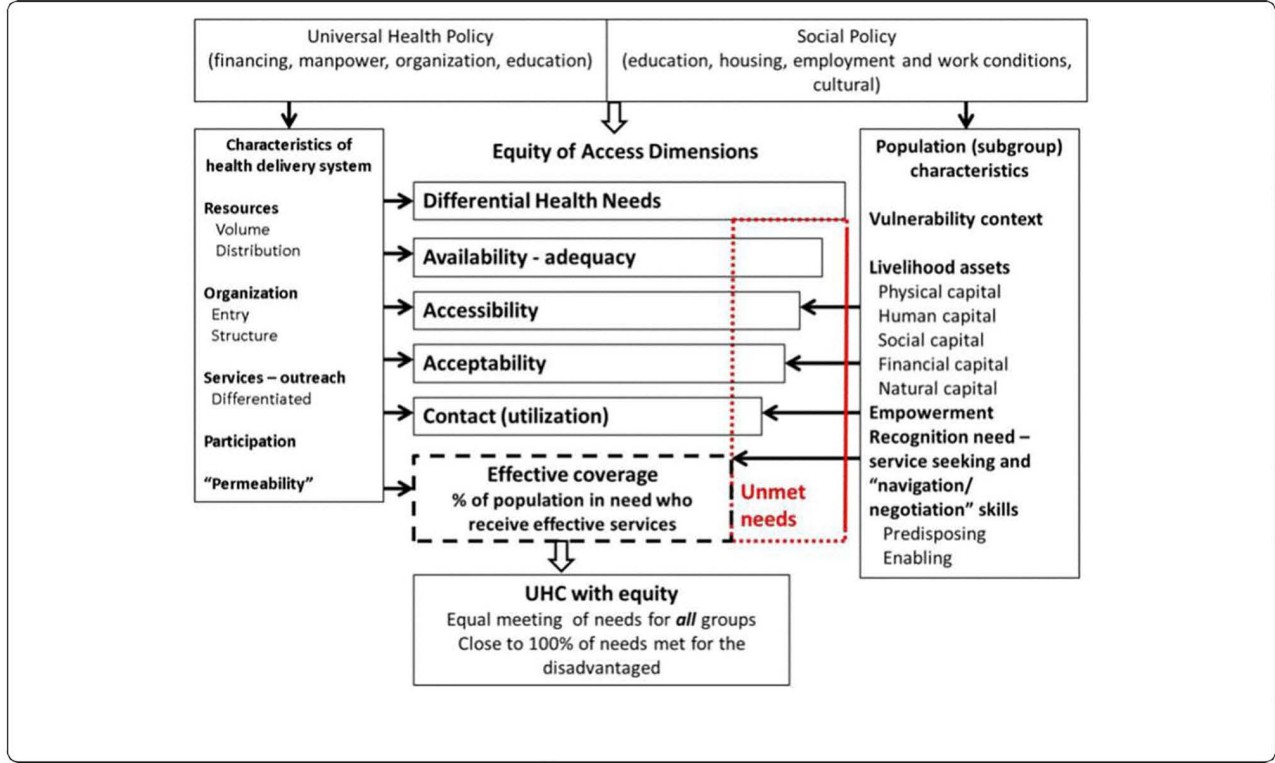

**Fig 1. Frame work of UHC and equity of access and effective coverage for all health needs as a result of supply and demand 'fit'. Source page 15, Frenz and Vega, 2010 [14,15].**

## Data source

This study utilized secondary data from the 2022 Uganda Demographic and Health Survey (UDHS). The survey employed a two-stage stratified sampling design to ensure national representativeness. In the first stage, 696 Enumeration Areas (EAs) were selected as primary sampling units with probability proportional to size, stratified by urban-rural residence within each of Uganda's 15 sub-regions. In the second stage, a fixed number of households were systematically selected from each EA. A total of 20,032 households were sampled, with interviews successfully completed in 19,588 occupied households, yielding a household response rate of 98.7%. Formal access to the anonymized dataset was obtained through written permission from the Uganda Bureau of Statistics (UBOS), the survey's implementing agency.

## Study population

The study population for the study was all women aged 15–49 years of age present in the household at the time of data collection.

## Inclusion criteria

The study included all women aged 15–49 years in the sample for the study

## Exclusion criteria

The study excluded women for whom data on the primary outcomes—cervical cancer screening utilization or awareness—were missing or incomplete, as these records could not contribute to the core analysis.

## Study variables and their measurement

**Independent variable.** The predictor for this study was socioeconomic status categorized using STATA 14 with the existing categorical variable v190 (the pre-constructed wealth quintile variable) divided into five wealth categories namely; Poorer, poor, middle, rich and richer constructed through Principal Component Analysis (PCA). This index synthesizes data on household ownership of durable assets (e.g., radio, television, refrigerator, bicycle, car), housing characteristics (e.g., materials used for floor, walls, and roof), and access to utilities and services (e.g., improved water source, sanitation facility type, main cooking fuel, electricity connection). Specific assets and characteristics included in the Uganda 2022 index are enumerated in the survey's final report. The PCA assigns weights to each variable based on their contribution to overall variance in living standards. The resulting continuous wealth scores for each household were then ranked nationally and divided into five equal groups, or wealth quintiles: Poorest, Poorer, Middle, Richer, Richest. This categorical variable (v190 in the dataset) serves as the primary indicator of socioeconomic status (SES) for equity analyses, enabling comparisons across distinct economic strata.

## Dependent variables

### (i) Utilization of cervical cancer screening

Screening utilization was assessed dichotomously based on a direct question in the Woman's Questionnaire administered to all eligible respondents aged 15–49 years. The specific question was: "Have you ever been screened for cervical cancer?" (Variable s119). Responses were recorded as 'Yes' or 'No'. For analytical purposes, a binary indicator was generated where a response of 'Yes' was coded as 1 (indicating utilization) and 'No' was coded as 0 (indicating non-utilization). This measure thus captures a self-reported lifetime history of ever having undergone any cervical cancer screening test. While it does not specify how recent or type of test, it provides a crucial baseline measure of service engagement.

### (ii) Awareness of Cervical Cancer Screening

Awareness was operationalized using a sequence of questions designed to gauge knowledge about cervical cancer and its prevention. The foundational question was: "Have you ever heard of any test for cervical cancer?" (Variable s100a). Women who answered affirmatively were then asked several follow-up questions probing their specific knowledge. For the core construct of screening awareness, the primary indicator used in this analysis was the response to the question: "Have you ever heard of any screening tests" that can tell if a woman has heard of any test for screening cervical cancer (Variable s100f). Responses to s100f ('Yes' or 'No') form a dichotomous variable directly measuring awareness of the existence of cervical cancer screening tests. Supplementary questions assessed broader knowledge dimensions, such as awareness of preventability ("Can cervical cancer be prevented?" s100e) and curability ("Can cervical cancer be cured?" s100d). However, the analysis of equity in screening awareness specifically focuses on responses to s100f as the most targeted measure.

## Statistical analysis

**Exploration of the data set.** The dataset exploration process focused on understanding its structure, contents, and relevance for analysis. This began with a thorough review of the data dictionary to comprehend variable definitions, coding schemes, and types, distinguishing between string and numeric variables. Where necessary, string variables were converted into numeric formats. The dataset was also checked for inconsistencies, duplicates, and missing values to ensure data integrity and assess its completeness. This detailed exploration established a robust foundation for conducting accurate and meaningful data analysis.

Data analysis was performed using the STATA 14 software (StataCorp, College Station, TX, USA). All variables included in this analysis were categorical in nature. Descriptive statistics for categorical variables, including demographic characteristics, screening utilization (s119), and screening awareness (s100f), were expressed as frequencies and per-centages (relative numbers).

The chi-square test was used to assess for statistically significant differences in the proportions of screening utilization and awareness across the categories of independent variables, such as residence and wealth quintile.

The Pearson chi-square ($\chi^2$) test was used to compare proportions between two independent groups, such as compar-ing screening rates between rural and urban residents. For analyses involving ordered categorical variables with more than two groups (specifically, wealth quintiles), the Cochran-Armitage trend $\chi^2$ test was employed to evaluate statistically significant trends across the ordered categories.

Concentration curves and indices were used to quantify socioeconomic-related inequality in cervical cancer screen-ing utilization and awareness, with interpretation guided by the principle of vertical equity. The concentration index (CI), bounded between −1 and +1, measures the degree of wealth-based inequality where a value of zero represents perfect equity. A negative CI indicates concentration of the outcome among poorer populations (pro-poor inequality) and the curve would be below the line of perfect equity, while a positive CI indicates concentration among wealthier populations (pro-rich inequality) and the curve would be above the line of perfect equity. For adverse outcomes like reduced screen-ing uptake, a negative CI demonstrates pro-rich inequity showing that poor women are disproportionately excluded from services despite likely greater need. For positive outcomes like screening awareness, a positive CI similarly indicates pro-rich inequity in knowledge distribution. The magnitude of the CI reflects the severity of inequality, with larger absolute values indicating stronger socioeconomic gradients. This approach directly tests whether health resources are distributed according to need rather than wealth, thereby assessing compliance with vertical equity principles. All statistical tests were two-sided. A p-value of less than 0.05 was considered statistically significant

## Results

### Summary of variables

The **Table 1** below shows a summary of study variables; Socio-economic status, Residence, utilization of cervical cancer screening and awareness of cervical cancer screening methods.

The study findings reveal significant demographic and geographic disparities in cervical cancer screening utilization among Ugandan women of reproductive age. From the total study population of 18,251 women, the majority (12,010

**Table 1. Summary of Study Variables.**

| Variable | Categories | Number | % |
|---|---|---|---|
| Socio-economic Status | Poorer | 3541 | 19.4 |
| | Poor | 3569 | 19.6 |
| | Middle | 3243 | 17.8 |
| | Rich | 3460 | 19 |
| | Richer | 4438 | 24.2 |
| Residence | Urban | 5,978 | 32.7 |
| | Rural | 12,270 | 67.3 |
| Cervical cancer Screening | Yes | 2,384 | 13 |
| | No | 15,867 | 87 |
| Awareness of cervical cancer screening | Yes | 8,088 | 58.4 |
| | No | 5764 | 41.6 |

women, 65.8%) resided in rural areas, while a smaller proportion (6,241 women, 35.2%) lived in urban settings, reflecting Uganda's predominantly rural population distribution. The overall cervical cancer screening utilization rate was notably low at 13%, indicating that only about one in eight women had accessed screening services. The overall prevalence of cervical cancer screening awareness among Ugandan women of reproductive age was 63%, indicating that nearly two-thirds of the population had received some information about the availability and importance of screening services

## Bivariable analysis

The analysis determined the relationship between soci0-economic statuses with utilization of cervical cancer screening and awareness among women of reproductive age in Uganda. The chi-square test determined the relationship while the Concentration Index evaluated the level of inequity as shown in the Table 2 below.

Bivariable analysis demonstrated a powerful socioeconomic gradient in both cervical cancer screening utilization and awareness in Uganda, with wealthier women consistently advantaged across both dimensions (Fig 2). The relationship between socioeconomic status and screening utilization was highly significant ($x^2 = 139.2$, $p < 0.000$), with a concentration index of 0.125 confirming substantial pro-rich inequality in service access. Similarly, socioeconomic status showed a strong association with screening awareness ($x^2 = 48.1$, $p < 0.000$), though the even higher concentration index of 0.178 ($p < 0.000$) revealed that awareness was even more disproportionately concentrated among affluent women than actual screening utilization.

This positive and statistically significant value is visually captured by a concentration curve (Lorenz curve) that bows distinctly below the line of perfect equality, demonstrating that the cumulative proportion of screening services is disproportionately concentrated among wealthier population segments. The pronounced curvature of the Lorenz curve indicates that women in the lowest wealth quintiles account for a far smaller share of screening utilization relative to their population size, while those in the highest quintiles utilize a disproportionately larger share. The curve also falls slightly below the perfect line for awareness of cervical cancer screening.

## Equity in cervical cancer screening utilization and awareness in Rural and Urban Uganda

The study findings Table 3 below show the equity in cervical cancer screening utilization and wareness among the urban and rural residents, the extent of association was measures usig the chi-square value and the level of inequity was measured using the concentraiton index.

Table 2.  Summary of Bi-variable Analysis.

| | Utilization | | Awareness | |
|---|---|---|---|---|
| | Yes | No | Yes | No |
| **Residence** | | | | |
| Rural | 1,436 | 10,834 | 5,030 | 4,022 |
| Urban | 947 | 5,031 | 3,057 | 1,741 |
| $x^2$ (P-value) | 62.3(P<0.000) | | 85.6(P<0.000) | |
| **Socio-economic Status** | | | | |
| Poorer | 299 | 2242 | 1285 | 929 |
| Poor | 399 | 3170 | 1537 | 1147 |
| Middle | 431 | 2812 | 1337 | 1153 |
| Rich | 514 | 2946 | 1625 | 1144 |
| Richer | 741 | 3699 | 2304 | 1391 |
| $x^2$ (P-value) | 139.2 (P<0.000) | | 48.1(P<0.000) | |
| CI (P-value) | 0.125(P<0.000) | | 0.178(P<0.000) | |

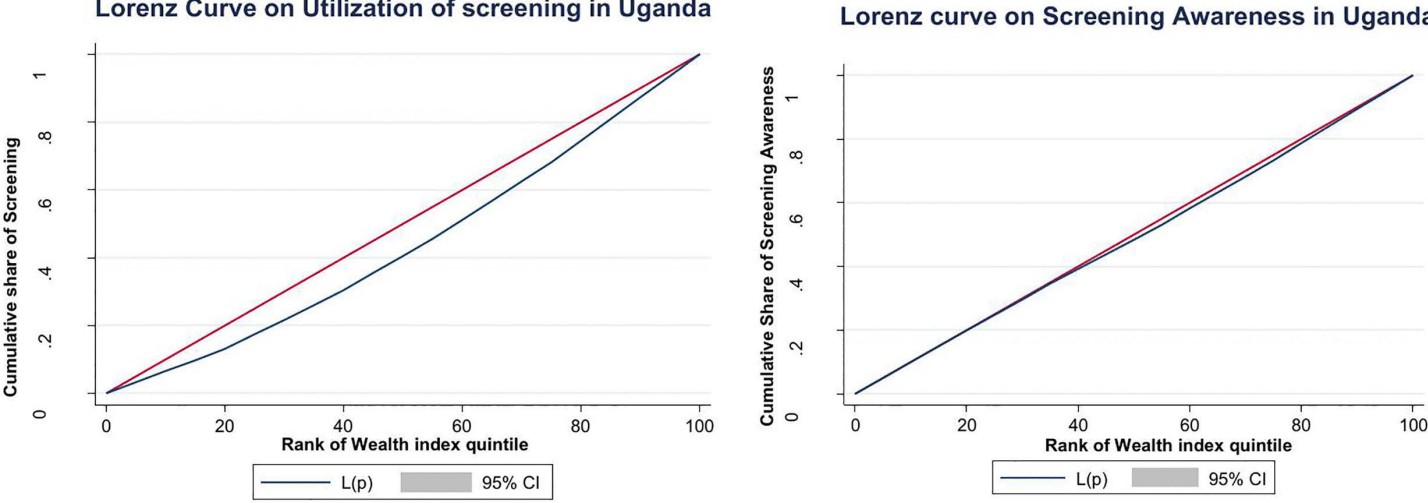

**Fig 2. Lorenz curves showing inequity in utilization and awareness of cervical cancer Screening in Uganda.**

**Table 3. Cervical cancer screening utilization and awareness in Rural and Urban Uganda.**

| | Rural | | Urban | |
|---|---|---|---|---|
| | Yes (%) | No (%) | Yes (%) | No (%) |
| **Utilization of Screening** | **1,436 (11.7)** | **10,834 (88.3)** | **944(15.8)** | **5,026 (84.20** |
| Poorer | 247 (17.2) | 2758 (25.5) | 52 (5.2) | 461 (8.8) |
| Poor | 325 (22.6) | 2761 (25.6) | 76 (7.7) | 421 (8.1) |
| Middle | 336 (23.4) | 2307 (21.5) | 102 (10.3) | 532 (10.1) |
| Rich | 366 (25.5) | 2089 (19.3) | 165 (16.7) | 925 (17.6) |
| Richer | 162 (11.3) | 879 (8.1) | 595 (60.1) | 2900 (55.4) |
| $\chi^2$ | 81.15 | P<0.000 | 16.68 | P<0.002 |
| CI | 0.049 | P<0.000 | 0.125 | P<0.000 |
| **Awareness of screening** | **5,030(55.6)** | **4022(44.4)** | **3,057(63.7)** | **1,747(36.4)** |
| Poorer | 1,866 (20.6) | 1,139 (35.7) | 210 (6.7) | 133 (7.1) |
| Poor | 2,327 (25.7) | 759 (23.8) | 220 (7) | 151 (8.3) |
| Middle | 2,046 (22.6) | 597 (18.7) | 274 (8.8) | 192 (10.4) |
| Rich | 1,980 (21.9) | 475 (14.9) | 532 (17) | 317 (17.2) |
| Richer | 825 (9.1) | 216 (6.8) | 1,890 (60.5) | 1,051 (57) |
| $\chi^2$ | 311.4 | P<0.000 | 8.19 | P<0.085 |
| CI | −0.007 | P>0.165 | 0.014 | P>0.016 |

## Utilization of cervical cancer screening

Despite the national screening rate of 13%, urban residents demonstrated higher screening utilization at 15.8%, while rural residents showed significantly lower utilization at 11.7%. The analysis revealed a pronounced pro-rich inequality in cervical cancer screening utilization in both urban and rural areas, though the disparity was markedly more severe in urban settings with the concentration index of 0.125 and the Lorenz curve (**Fig 3**) far away from the line of equality

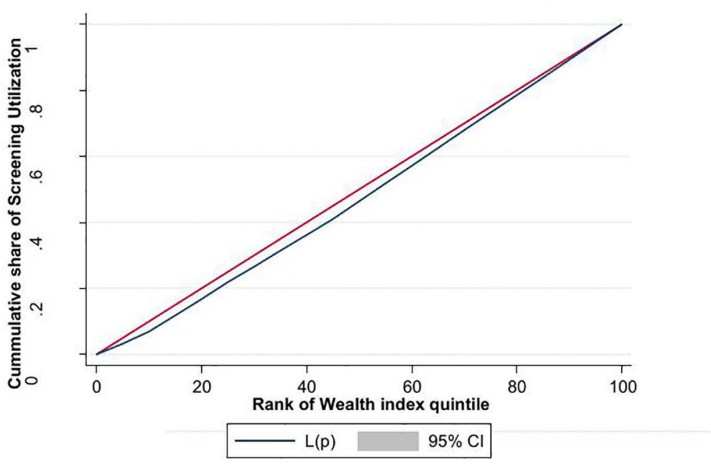
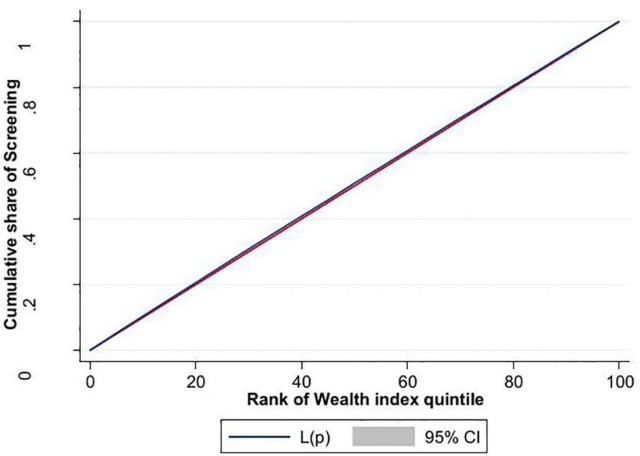

**Fig 3. Lorenz curve for Utilization of cervical cancer screening among Urban and Rural.**

indicated that access to screening was strongly concentrated among wealthier women, reflecting significant financial and structural barriers that exclude poorer urban residents from services. In rural areas, the concentration index of 0.049 and the curve slightly below the line of perfect equity also demonstrated a pro-rich distribution, but with comparatively milder inequality, suggesting that while socioeconomic status still influenced access, the geographic and resource limitations in rural regions may partially equalize disparities across wealth quintiles. Both values were statistically significant ($p < 0.000$), confirming that socioeconomic status is a powerful determinant of screening utilization. These findings highlight a systemic failure to achieve vertical equity, as the distribution of services disproportionately benefits affluent populations rather than being allocated based on the greater need associated with lower socioeconomic status.

### Cervical cancer screening awareness

The analysis of equity in awareness of cervical cancer screening revealed divergent patterns between rural and urban populations as evidenced by the concentration indicies and Lorenz curves in **Fig 4**. Among rural residents, the concentration index of −0.007 ($P > 0.165$) indicated a negligible and statistically non-significant pro-poor distribution, suggesting that awareness was nearly equitably distributed across wealth quintiles in these areas. The corresponding concentration curve would lie very close to the line of perfect equality, showing only a minimal tendency for awareness to be slightly more concentrated among poorer groups. In contrast, urban areas demonstrated a concentration index of 0.014 ($P < 0.016$), reflecting a statistically significant pro-rich inequality where knowledge of screening was disproportionately concentrated among wealthier women. The urban concentration curve would bow slightly below the equity line, visually confirming that the cumulative share of awareness among lower wealth quintiles falls short of their population proportion. While both indices reflect relatively mild inequality in absolute terms, the statistically significant pro-rich distribution in urban settings highlights how systemic factors potentially including differential access to health information, education, and media exposure create subtle but meaningful advantages for affluent urban women in acquiring health knowledge, even as rural areas achieve near-equitable awareness distribution despite their broader resource constraints

## Discussion

This study provides evidence of significant socioeconomic and geographic inequities in cervical cancer screening awareness and utilization among Ugandan women of reproductive age. The findings reveal a consistent pro-rich gradient in both

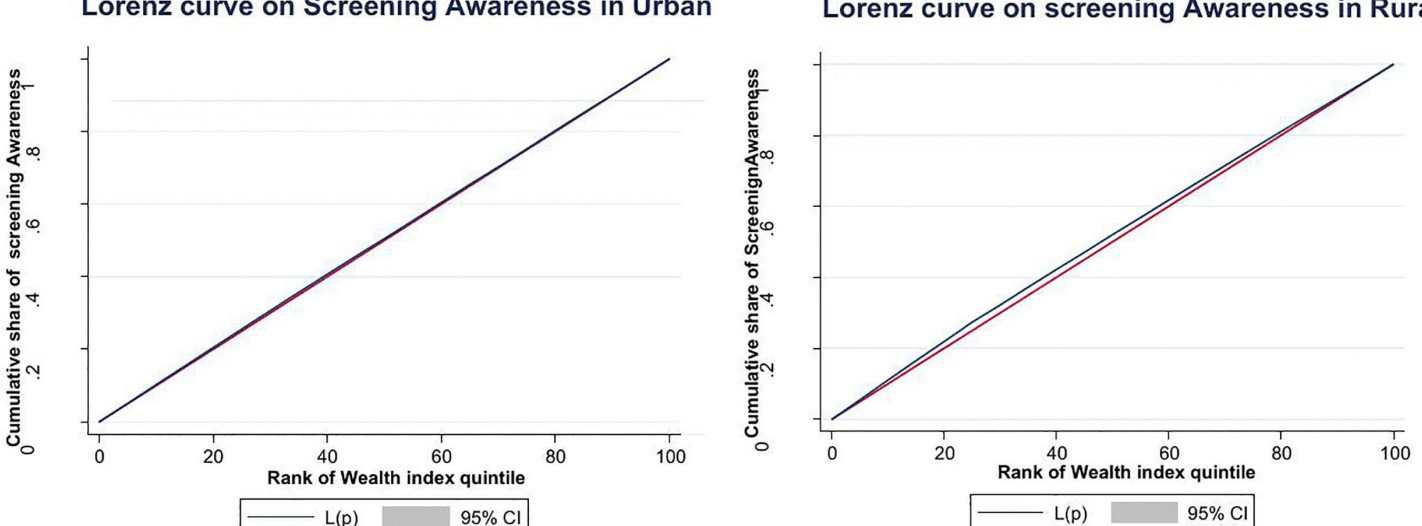

**Fig 4. Lorez curve indicating the level of inequity of cervical cancer screnign awareness among urban and rural residents.**

dimensions of screening engagement, with wealthier women demonstrating substantially greater awareness of screening services and higher utilization rates compared to their poorer counterparts which is consistent with earlier studies documenting equity in cervical cancer screening [16,17]. The concentration index of 0.178 for awareness and 0.125 for utilization, both statistically significant, strongly affirms how socioeconomic status operates as a powerful determinant of preventive health behaviors, creating a double disadvantage for women in lower wealth quintiles who are both less informed about and less able to access potentially life-saving screening services.

The significantly higher concentration index for awareness (0.178) compared to utilization (0.125) highlights that information inequity as a primary barrier [18]. This finding directly implies that awareness campaigns must be specifically targeted to reach poorer and rural women to address this foundational inequity. The urban-rural disparities further compound these inequities, with urban women demonstrating both higher awareness (64% vs. 58%) and utilization (15.8% vs. 11.6%) than their rural counterparts. These geographic differences are indicative of broader structural inequities in health infrastructure, resource allocation, and information dissemination. Consequently, programmatic efforts to increase screening must explicitly prioritize rural health system strengthening and the deployment of mobile or community-based services [19].

Our findings align with and extend previous research on health inequities in low-income settings. The pro-rich distribution of screening services contradicts the fundamental principle of vertical equity, which requires that health resources be allocated according to need rather than ability to pay [20]. This is particularly concerning given evidence that rural and low-income women in Uganda may have higher prevalence of HPV infection and other risk factors, suggesting that those with greatest biological need have least access to preventive services [21]. The persistence of these inequities alongside Uganda's commitment to universal health coverage is associated with substantial implementation gaps. To fulfill UHC promises, policy must explicitly redress these gaps by reorienting cervical cancer prevention strategies through an equity lens

## Study strengths and limitations

The study's limitations include its cross-sectional design, which prevents causal inference, and the reliance on self-reported data for screening behavior, which may be subject to recall bias. Additionally, while the wealth index provides a valuable composite measure of socioeconomic status, it may not capture all dimensions of economic vulnerability that

influence health-seeking behavior. Nevertheless, the large, nationally representative sample and use of standardized DHS methodology strengthen the validity and generalizability of our findings.

## Conclusion and policy implications

These results have urgent policy implications for cervical cancer control in Uganda and similar settings. Given the steeper pro-rich gradient in awareness than in utilization, the first policy imperative is to launch targeted awareness campaigns using appropriate channels to reach poor and rural women. Simultaneously, and informed by the pro-rich inequality in utilization itself, interventions must address supply-side (access) barriers with the same equity focus. Corresponding to the finding that wealth is a strong determinant of utilization, financial protection mechanisms must be strengthened to eliminate out-of-pocket payments, which disproportionately deter poorer women from seeking screening. Finally, to directly counteract the urban-rural utilization disparity, resource allocation formulas must explicitly incorporate equity criteria to ensure screening services are preferentially located in areas of greatest need, such as rural regions, rather than areas of greatest wealth

Addressing these inequities is not merely an ethical imperative but a practical necessity for achieving meaningful progress in cervical cancer control. Only through deliberately equitable policies and programs can Uganda hope to reduce its disproportionate burden of cervical cancer mortality and move toward genuine universal health coverage.

## Author contributions

**Conceptualization:** Geofrey Emesu.

**Formal analysis:** Geofrey Emesu.

**Methodology:** Geofrey Emesu, Elizabeth Ekirapa Kiracho, Alfred Jatho.

**Resources:** Annette Kyomuhangi.

**Software:** Aggrey David Mukose.

**Supervision:** Elizabeth Ekirapa Kiracho, Alfred Jatho, Joseph Kagaayi.

**Writing – original draft:** Geofrey Emesu.

**Writing – review & editing:** Geofrey Emesu.

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
