## [Decision Letter · Decision Letter 0]

26 Jan 2026

Dear Dr. Emesu,

Thank you for submitting your manuscript to PLOS ONE. After careful consideration, we feel that it has merit but does not fully meet PLOS ONE’s publication criteria as it currently stands. Therefore, we invite you to submit a revised version of the manuscript that addresses the points raised during the review process.

We look forward to receiving your revised manuscript.

Kind regards,

Aloysius Gonzaga Mubuuke

Academic Editor

PLOS One

Journal Requirements:

2. In this instance it seems there may be acceptable restrictions in place that prevent the public sharing of your minimal data. However, in line with our goal of ensuring long-term data availability to all interested researchers, PLOS’ Data Policy states that authors cannot be the sole named individuals responsible for ensuring data access (http://journals.plos.org/plosone/s/data-availability#loc-acceptable-data-sharing-methods).

3. If any figure files for review show as item type ‘other’ please change to item type ‘figure’ as the reviewer does not have access to these ’other’ files.

Additional Editor Comments:

As you address the review comments, also proof-read the entire paper to correct language errors.

Reviewers' comments:

Reviewer's Responses to Questions

**Comments to the Author**

1. Is the manuscript technically sound, and do the data support the conclusions?

Reviewer #1: Yes

Reviewer #2: Yes

2. Has the statistical analysis been performed appropriately and rigorously?

Reviewer #1: Yes

Reviewer #2: Yes

3. Have the authors made all data underlying the findings in their manuscript fully available?

Reviewer #1: No

Reviewer #2: Yes

4. Is the manuscript presented in an intelligible fashion and written in standard English?

Reviewer #1: Yes

Reviewer #2: Yes

Reviewer #1: Dear authors thank you for submitting the manuscript. This is very well written manuscript and the employed methodology is rigor to answer the research question. To further improve the work the following are my observations and suggestions

Title: Inclusion of Urban and Rural is kind of skeptical because, it's very much known that Uganda has both. I would suggest omission of the words and consider retaining Uganda alone

Abstract

Background: While articulating the background information withe objectives, the objective should be written in past tense

Methods: It's important to state at what level was the p-value set to report significance

Results: Here the sentence "National screening utilization was 13%" sounds ectopic because this wasn't part of the objectives and hence shouldn't be reported and as the matter of fact it's reported in the UDHS report

Conclusion: You have reported that "There are stark socioeconomic and geographic inequities in cervical cancer screening in Uganda" Was this a primary goal of your research? I suggest conclusion of the key findings mirroring the study research question

Background: While it's very well written, I suggest the inclusion of the consequences of not screening for cervical cancer

Methodology:

Study design: It important to make sure that it's explicitly study design as it's observed that, there more unnecessary information included. The definition of Equity and inequity shouldn't be one this sub-section and wherever appropriately placed it's source should be cited

Description of data set: This should be rephrased to Data source where you will also need to include sampling procedures undertaken

Exploration of dataset: This subsection should be in the analysis plan

Exclusion criteria is not the opposite of the inclusion criteria please revise

Independent variables: Make sure that you have operationalize the variables by including the how they were coded in STATA and why so ?

The categorization and recategorization of the variables based on what grounds ?

Dependent variables: The word utilization and use are very much different please revisit oxford dictionary and choose the appropriate word.

Statistical Analysis: The sentence "Associations between categorical variables were assessed using chi-square tests" is not right. Refer to the book titled "Essential Medical Statistics" written by Betty Kirkwood where its clearly stated that chi-square can not assess the association but rather the distribution of independent variable to the dependent variable

RESULTS: Well written however:

As the rule of thumb do you think its not important to describe your study population as objective zero?

DISCUSSION: Its also important to discuss the implication of the results as well as comparison from other findings

Reviewer #2: This manuscript examines socioeconomic and geographic inequities in cervical cancer screening awareness and utilization among women of reproductive age in Uganda using data from the 2022 Uganda Demographic and Health Survey. The topic is highly relevant and timely, particularly in the context of the World Health Organization’s cervical cancer elimination strategy and universal health coverage commitments. The manuscript is generally intelligible and logically structured, with clearly stated objectives and an appropriate conceptual grounding in vertical equity. The authors demonstrate strong familiarity with the policy and epidemiological context and present findings that are broadly consistent with existing evidence from low- and middle-income settings. However, while the scientific intent and overall structure are sound, the manuscript requires substantial revision before it can be considered for publication. The language, although understandable, contains numerous typographical, grammatical, and stylistic errors that reduce clarity and readability. These include repeated spelling mistakes, inconsistent punctuation and spacing, overly long or fragmented sentences, and inconsistent terminology across sections. In addition, several numerical values, percentages, and denominators are reported inconsistently between the text and tables, which undermines interpretability. As PLOS ONE does not copyedit accepted manuscripts, careful language editing and harmonisation of all reported figures are essential at revision.

The study relies on secondary data from the 2022 Uganda Demographic and Health Survey, obtained with permission from the Uganda Bureau of Statistics, which is appropriate and ethically sound. However, the data availability statement should be strengthened to fully comply with PLOS ONE policy. The manuscript should explicitly state that the data are owned by a third party and are publicly accessible upon request through the DHS Program or UBOS, with clear instructions provided for access. While the authors cannot share raw data directly, they should clarify that all analyses can be replicated using the publicly available dataset. If analytic code or derived datasets were generated, the authors are encouraged to deposit these in a public repository or state that they are available upon reasonable request. The statistical approach is broadly appropriate for the research objectives and data source, and the use of descriptive statistics, chi-square tests, and concentration indices to examine socioeconomic and rural–urban inequities aligns well with the stated vertical equity framework. Nevertheless, several methodological clarifications are required. The authors should explicitly state whether sampling weights, clustering, and stratification were applied in all analyses, including inequality measures, as this is essential for nationally representative inference. Given that the primary outcomes are binary, the manuscript should specify whether a corrected concentration index (such as the Erreygers or Wagstaff correction) was used and justify the choice. Confidence intervals for key estimates, including prevalence and concentration indices, are not consistently reported and should be added. In addition, internal inconsistencies in sample sizes and percentages across sections must be resolved, and interpretation of concentration indices should be standardised with consistent use of “pro-rich” and “pro-poor” terminology.

Overall, the manuscript presents a technically sound observational analysis, and the data largely support the authors’ conclusions regarding low screening uptake and persistent inequities in cervical cancer screening awareness and utilization in Uganda. The findings clearly demonstrate that wealthier women are disproportionately advantaged in both knowledge of and access to screening services, and the discussion appropriately situates these results within broader debates on health equity and universal health coverage. The policy implications proposed, including the need for targeted awareness interventions, equity-sensitive resource allocation, and financial protection mechanisms, are reasonable and broadly supported by the data. However, the authors should strengthen the linkage between specific quantitative findings and their policy recommendations, ensure complete internal consistency across results, and avoid causal language given the cross-sectional design. There are no apparent concerns regarding dual publication, and the use of anonymized DHS data is ethically appropriate, although an explicit ethics statement confirming secondary-data approval or exemption should be included. In summary, this is a relevant and potentially valuable contribution to the literature that would be suitable for publication in PLOS ONE after major revisions addressing language quality, methodological transparency, data availability clarification, and consistency of reporting.

**Do you want your identity to be public for this peer review?** For information about this choice, including consent withdrawal, please see our For information about this choice, including consent withdrawal, please see our Privacy Policy .

Reviewer #1: No

Reviewer #2: **Yes:** Edward Mawejje, MD, MPH, IMPSEdward Mawejje, MD, MPH, IMPS

---

## [Author Response · Author response to Decision Letter 1]

4 Feb 2026

We wish to extend our sincere gratitude to both reviewers for their thoughtful, detailed, and constructive critiques of our manuscript. Your combined expertise has been invaluable in significantly strengthening this work.

Your comments collectively guided essential improvements across several key dimensions: the imperative for professional language editing and internal consistency, the need for greater methodological transparency and rigor (particularly regarding survey weights and concentration indices), and the importance of a more impactful discussion that clearly links findings to implications and policy.

We have addressed each point conscientiously. The revised manuscript now features polished language and consistent reporting, explicit methodological details compliant with best practices for complex survey analysis, and a substantially reframed discussion that directly translates specific quantitative inequities into actionable insights for cervical cancer prevention policy in Uganda.

We are confident that your guidance has transformed this into a more robust, clear, and valuable contribution to the literature. Thank you for your time and for helping to improve our research.

---

## [Decision Letter · Decision Letter 1]

24 Mar 2026

Equity in  Awareness and Utilization of Cervical Cancer Screening Services among Women of Reproductive Age in Uganda: Analysis of vertical Equity using evidence from UDHS 2022

PONE-D-25-54550R1

Dear Dr. Emesu,

We’re pleased to inform you that your manuscript has been judged scientifically suitable for publication and will be formally accepted for publication once it meets all outstanding technical requirements.

Kind regards,

Aloysius Gonzaga Mubuuke

Academic Editor

PLOS One

Additional Editor Comments (optional):

Reviewers' comments:

Reviewer's Responses to Questions

**Comments to the Author**

Reviewer #1: All comments have been addressed

Reviewer #2: All comments have been addressed

2. Is the manuscript technically sound, and do the data support the conclusions?

Reviewer #1: Yes

Reviewer #2: Yes

3. Has the statistical analysis been performed appropriately and rigorously?

Reviewer #1: Yes

Reviewer #2: Yes

4. Have the authors made all data underlying the findings in their manuscript fully available?

Reviewer #1: Yes

Reviewer #2: Yes

5. Is the manuscript presented in an intelligible fashion and written in standard English?

Reviewer #1: Yes

Reviewer #2: Yes

Reviewer #1: Thank you for reviewing the manuscript. It has take it shape now and you have addressed all comments I raised and hence I recommend publication. Thank you and best wishes

Reviewer #2: Article well revized and fit for considerations nased on the revisions docuemented, well anotated and classical solutions been inserted at last.

**Do you want your identity to be public for this peer review?** For information about this choice, including consent withdrawal, please see our For information about this choice, including consent withdrawal, please see our Privacy Policy .

Reviewer #1: No

Reviewer #2: **Yes:** Dr Edward Mawejje, MD, MPH, IMPSDr Edward Mawejje, MD, MPH, IMPS

---

## [Editor Report · Acceptance letter]

PONE-D-25-54550R1

PLOS One

Dear Dr. Emesu,

I'm pleased to inform you that your manuscript has been deemed suitable for publication in PLOS One. Congratulations! Your manuscript is now being handed over to our production team.

Kind regards,

on behalf of

Dr. Aloysius Gonzaga Mubuuke

Academic Editor

PLOS One